# Antioxidant Activity and Mechanism of Action of Amwaprin: A Protein in Honeybee (*Apis mellifera*) Venom

**DOI:** 10.3390/antiox13040469

**Published:** 2024-04-17

**Authors:** Bo-Yeon Kim, Kwang-Sik Lee, Byung-Rae Jin

**Affiliations:** College of Natural Resources and Life Science, Dong-A University, Busan 49315, Republic of Korea; boyeon@dau.ac.kr

**Keywords:** *Apis mellifera*, honeybee, antioxidant agent, venom, waprin

## Abstract

Bee venom contains several bioactive components, including enzymatic and non-enzymatic proteins. There is increasing interest in the bioactive components of bee venom since they have exhibited various pharmacological effects. Recently, *Apis mellifera* waprin (Amwaprin) was identified as a novel protein in *Apis mellifera* (honeybee) venom and characterized as an antimicrobial agent. Herein, the novel biological function of Amwaprin as an antioxidant is described. In addition, the antioxidant effects of Amwaprin in mammalian cells were investigated. Amwaprin inhibited the growth of, oxidative stress-induced cytotoxicity, and inflammatory response in mammalian NIH-3T3 cells. Amwaprin decreased caspase-3 activity during oxidative stress and exhibited protective activity against oxidative stress-induced cell apoptosis in NIH-3T3 and insect Sf9 cells. The mechanism underlying the cell protective effect of Amwaprin against oxidative stress is due to its direct binding to the cell membrane. Furthermore, Amwaprin demonstrated radical-scavenging activity and protected against oxidative DNA damage. These results suggest that the antioxidant capacity of Amwaprin is attributed to the synergistic effects of its radical-scavenging action and cell shielding, indicating its novel role as an antioxidant agent.

## 1. Introduction

*Apis mellifera* (honeybee) venom has long been utilized as a traditional medicine and contains numerous bioactive components, including enzymatic and non-enzymatic proteins [1,2,3]. Thus, honeybee venom is gaining attention for its potential role in drug development to treat various diseases, such as arthritis, rheumatism, pain, tumors, and skin disease [4,5,6,7]. Various peptides have been identified in honeybee venom, including melittin, apamin, adolapin, secapin, mast cell degranulating peptide, and protease inhibitors [2,4,5,8,9,10]. Melittin, an amphiphilic peptide consisting of 26 amino acid residues, is a major component of honeybee venom (approximately 50% of its dry weight), and in addition to its toxic properties, it has several biological activities, including anti-inflammatory and anti-cancer effects [4,5,10,11]. Secapin is a neurotoxin in honeybee venom [12] but also exhibits multifunctional activities, such as anti-fibrinolytic, anti-elastolytic, and anti-microbial activities [9]. Several protease inhibitors function as anti-fibrinolytic and anti-microbial agents [8,13]. As such, honeybee venom components have been widely explored as natural agents for potential alternative uses, such as anti-microbial, antioxidant, anti-cancer, and anti-inflammatory activities [14,15,16,17].

Since the peptide components of honeybee venom exhibit various biological activities, current research is focused on identifying the novel peptide components and exploring their functions. Recently, we identified *Apis mellifera* waprin (Amwaprin) as a novel honeybee venom protein that acts as an antimicrobial agent against microbial serine proteases and elastases while having no hemolytic activity [18], which is an important safety parameter for potential therapeutic agents [19,20]. Waprins, which contain a whey-acidic protein (WAP) four-disulfide core domain [21], have been identified as proteins in the venom of lower vertebrates and invertebrates, such as snakes, frogs, scorpions, ants, wasps, and honeybees [18,22,23,24,25,26,27]. Although waprins from snakes, frogs, and honeybees have exhibited antimicrobial activities against bacteria and fungi [18,22,24], their functional characterization remains limited.

In addition to the identification of new proteins, such as Amwaprin and superoxide dismutase [18,28], novel functions of known honeybee venom components, including secapin, serine protease, protease inhibitors, and major royal jelly proteins 8–9, have also been characterized [8,9,13,29,30]. Current research on honeybee venom is directed toward its pharmacological and toxicological effects [4,5,6,10,11,28,30]. For instance, despite a previous study demonstrating significant anti-cancer activity of melittin against malignant melanoma, melittin is widely known for its strong hemolytic activity [11]. To develop effective pharmacological components, research needs to discover less toxic and more effective or multifunctional natural agents [14]. Considering that several honeybee venom proteins possess multifunctional properties, an in-depth understanding of the mechanism of action for these effects is crucial.

This study presents the first evidence of the antioxidant activity of Amwaprin and its mechanism of action in mammalian cells. This study reveals that Amwaprin protects mammalian cells against oxidative stress-induced cytotoxicity and inflammatory responses. Furthermore, this study demonstrates that Amwaprin decreases caspase-3 activity during oxidative stress, and its cell protective effect against oxidative stress-induced cell apoptosis is due to its direct binding to the cell membrane. Lastly, this study reveals that Amwaprin exhibits antioxidative and scavenging activities against reactive oxygen species (ROS). Thus, these findings demonstrate the novel function of Amwaprin as an antioxidant agent.

## 2. Materials and Methods

### 2.1. Recombinant Amwaprin and Polyclonal Antibodies

The recombinant Amwaprin protein and anti-Amwaprin antibody produced in our previous study [18] were used. Recombinant Amwaprin was produced in baculovirus-infected insect cells using a baculovirus expression vector system. The insect cell line *Spodoptera frugiperda* 9 (Sf9; Gibco BRL, Gaithersburg, MD, USA) was cultured in TC100 medium (Gibco BRL) supplemented with 10% fetal bovine serum (FBS, Gibco BRL) at 27 °C. *Amwaprin* cDNA was inserted into the baculovirus expression vector *pBacPAK8* (Clontech, Palo Alto, CA, USA). Recombinant Amwaprin baculoviruses were infected into Sf9 cells, and the recombinant Amwaprin, including a hexahistidine tag (His-tag), was purified using a MagneHis^TM^ Protein Purification System (Promega, Madison, WI, USA). The recombinant Amwaprin was quantified using a protein assay kit (Bio-Rad, Hercules, CA, USA). An anti-Amwaprin antibody against the recombinant Amwaprin protein generated in BALB/c mice (Samtako Bio Korea Co., Osan, Republic of Korea) was used.

### 2.2. Cell Culture and Cell Growth Assays

This study used the murine fibroblast cell line NIH-3T3 (AddexBio, San Diego, CA, USA) to assess the cellular response to recombinant Amwaprin. NIH-3T3 cells were cultured in Dulbecco’s modified Eagle’s medium (DMEM, Sigma-Aldrich, St. Louis, MO, USA) supplemented with 10% FBS (Gibco BRL) at 37 °C in a humidified incubator containing 5% CO_2_. The cells were counted using a hemocytometer. The cell growth assessment of the NIH-3T3 cells was performed utilizing an MTT (3-(4,5 dimethyl-thiazol-2-yl)-2,5-diphenyl tetrazolium bromide) Cell Proliferation Assay Kit (BioVision, Milpitas, CA, USA). NIH-3T3 cells (2 × 10^4^ cells/well of a 96-well plate) were incubated with serial dilutions of recombinant Amwaprin (0, 50, 250, or 500 ng/mL) or 1% Triton X-100 (Sigma-Aldrich) as a positive control that inhibits cell proliferation. Cells without Amwaprin and/or Triton X-100 were considered the negative control. The treated cells were incubated for 24 or 48 h, followed by the addition of 50 μL of the MTT reagent (BioVision) to each well. Following a 4 h incubation, the media was removed, and the formazan crystals were dissolved in 150 μL of MTT solvent. Absorbance was measured at 590 nm using a microplate reader (Infinite F50 Model, Tecan, Grödig, Austria). Additionally, the number of NIH-3T3 cells with or without recombinant Amwaprin (250 ng/mL) was counted for five days.

### 2.3. Reactive Oxygen Species (ROS) Assay

NIH-3T3 cells (2 × 10^4^ cells/well of a 6-well plate) were incubated with or without 250 ng/mL recombinant Amwaprin for five days. The media was harvested daily, and the ROS levels were quantified using an OxiSelect^TM^ In Vitro ROS/RNS Assay Kit (Green Fluorescence; Cell Biolabs, Inc., San Diego, CA, USA) according to the manufacturer’s instructions. In this reaction, ROS and RNS species were reacted with a fluorogenic probe, 2′,7′-dichlorodihydrofluorescein, under dark conditions at room temperature for 30 min. The fluorescence of each well was measured at 480 nm excitation/530 nm emission using a microplate reader (Infinite F50 Model, Tecan).

### 2.4. Cytotoxicity Assay

NIH-3T3 cells were seeded in 96-well plates (2 × 10^4^ cells/well) for 24 h. Subsequently, the cells were incubated with or without 250 ng/mL recombinant Amwaprin or H_2_O_2_ (50 μM, Sigma-Aldrich) for 24 or 48 h. The media was harvested via centrifugation at 1000× *g* for 5 min and directly used in the lactate dehydrogenase (LDH) assay using an LDH Cytotoxicity Assay Kit II (BioVision). The absorbance was measured at 405 nm using a microplate reader (Infinite F50 Model, Tecan).

### 2.5. Apoptosis Assay

The apoptotic response to recombinant Amwaprin in NIH-3T3 cells was determined by measuring the caspase-3 activity using a Caspase-3/CPP32 Colorimetric Assay Kit (BioVision). NIH-3T3 cells were cultured in 96-well plates (2 × 10^4^ cells/well) and treated with 250 ng/mL recombinant Amwaprin and 50 μM H_2_O_2_ for 24 or 48 h. After treatment, the cells were washed with phosphate-buffered saline (PBS, 140 mM NaCl, 27 mM KCl, 8 mM Na_2_HPO_4_, and 1.5 mM KH_2_PO_4_; pH 7.4) and resuspended in cold lysis buffer. After centrifugation at 14,000× *g* for 15 min, the supernatants were mixed with a caspase substrate (Ac-DEVD-pNA) in a 96-well plate and incubated overnight at 37 °C. The released *p*-nitroaniline levels were determined by measuring the absorbance at 405 nm.

### 2.6. Enzyme-Linked Immunosorbent Assay (ELISA)

The levels of interleukin (IL)-1β, IL-6, and tumor necrosis factor (TNF)-α were determined using ELISA Kits (Abcam, Cambridge, UK). NIH-3T3 cells were cultured in 96-well plates (2 × 10^4^ cells/well) and treated with 250 ng/mL recombinant Amwaprin and 50 μM H_2_O_2_ for 24 or 48 h. The ELISA assays were performed as described in our previous study [28]. Briefly, the media from the treated NIH-3T3 cells was collected and added to each well of a 96-well plate coated with an anti-mouse IL-1β, IL-6, or TNF-α antibody (Abcam). Following incubation, the wells were washed four times with 300 μL of the washing buffer. Subsequently, 100 μL of a biotinylated anti-mouse IL-1β, IL-6, or TNF-α antibody was added to each well and incubated for 1 h at room temperature. The wells were repeatedly washed, and 100 μL of the TMB One-Step Substrate Reagent was added to each well and incubated for 30 min, followed by the addition of 50 μL of the stop solution. The absorbance was determined at 450 nm using a microplate reader (Infinite F50 Model, Tecan).

### 2.7. Immunofluorescence Staining

The NIH-3T3 and Sf9 cells were incubated with or without 250 ng/mL recombinant Amwaprin for 24 h. The cell samples were fixed with acetone (−20 °C) for 2 min, washed with PBS, and then incubated with 2% bovine serum albumin (BSA) at room temperature for 20 min. Subsequently, the cell samples were washed with PBS and incubated with the mouse anti-Amwaprin antibody, diluted 1:400 (*v*/*v*). Then, the cell samples were incubated with the tetramethyl rhodamine isothiocyanate-conjugated goat anti-mouse secondary antibody, diluted 1:400 (*v*/*v*) (Santa Cruz Biotech, Inc., Santa Cruz, CA, USA). Images of recombinant Amwaprin bound to the cell membrane were obtained using laser-scanning confocal microscopy (Carl Zeiss LSM 510, Zeiss, Jena, Germany).

### 2.8. Apoptotic Cell Death Imaging

NIH-3T3 and Sf9 cells were treated with 250 ng/mL recombinant Amwaprin in the presence of 50 μM H_2_O_2_ for 24 h. The cells were double-labeled with an in situ Cell Death Detection Kit (Roche Applied Science, Mannheim, Germany) and a mouse anti-Amwaprin antibody. Following the treatment, NIH-3T3 and Sf9 cells were washed thrice with PBS and preincubated with 5% BSA in PBS for 1 h. The cell samples were then incubated with the anti-Amwaprin antibody diluted 1:400 (*v*/*v*) in PBS containing 1% BSA for 1 h, followed by incubation with the tetramethyl rhodamine isothiocyanate-conjugated goat anti-mouse IgG antibody diluted 1:400 (*v*/*v*) (Santa Cruz Biotech, Inc.) in PBS containing 1% BSA for 3 h. The cells were washed thrice with PBS and treated with a TUNEL reaction mixture containing terminal deoxynucleotidyl transferase and fluorescein-conjugated deoxyuridine triphosphatase at 37 °C for 1 h. Finally, the cell samples were washed with PBS, wet mounted, and the presence of Amwaprin and apoptosis was observed for each treatment. Images were obtained using laser-scanning confocal microscopy (Carl Zeiss LSM 510).

### 2.9. Radical-Scavenging Assay

The free radical scavenging activity of recombinant Amwaprin was determined using 1,1-diphenyl-2-picrylhydrazyl (DPPH). Samples (100 μL) of recombinant Amwaprin (0, 250, 500, 1000, or 3000 ng) were treated with 200 μL of the DPPH radical solution (200 μM) and 70 μL of ethanol (99%, Sigma-Aldrich) for 30 min. After centrifugation at 16,000× *g* for 5 min, the absorbance of the supernatant was measured at 517 nm. The residual radical (%) of DPPH was determined using the equation % = (absorbance of the Amwaprin sample)/(absorbance of the control) × 100 [31].

### 2.10. In Vitro DNA Protection Assay

A metal-catalyzed oxidation (MCO) DNA cleavage protection assay was performed using reaction mixtures (total volume of 50 μL) containing 16.5 μM FeCl_3_ and 3.3 mM dithiothreitol (DTT) in the presence of recombinant Amwaprin. pUC18 super-coiled plasmid DNA (1 μg) was incubated with or without recombinant Amwaprin (50, 250, or 500 ng) at 37 °C for 1.5 h and then treated with the reaction mixture at 37 °C for 1.5 h. DNA nicking was observed by 1.0% agarose gel electrophoresis.

### 2.11. Statistical Analysis

Statistical analysis was conducted using IBM PASW^®^ Statistics 22.0 (IBM Inc., Chicago, IL, USA). Data are expressed as the mean ± standard deviation (SD). We performed a Shapiro–Wilk normality test followed by Tukey’s honestly significant difference post hoc test for multiple comparisons. For comparisons of two samples, we performed a Wilcoxon Rank Sum Test followed by a Two-Sample *t*-test. *p*-values are denoted as follows: *, *p* < 0.05; **, *p* < 0.01; and ***, *p* < 0.001.

## 3. Results

### 3.1. Amwaprin Inhibits Cell Growth

In our previous study, we identified and characterized Amwaprin, which exhibits microbicidal and anti-elastolytic activities, along with being heat-stable and non-hemolytic [18]. Amwaprin exhibits characteristic features of the waprin family, which includes a WAP domain with eight conserved cysteine residues (Figure 1). Unlike waprins from snake venoms [22] and frog skins [24], Amwaprin has an additional four conserved cysteine residues following the WAP domain.

To determine whether Amwaprin affects cellular growth, NIH-3T3 cells were treated with recombinant Amwaprin. The MTT assay revealed that, compared with the untreated cells at 24 and 48 h, the NIH-3T3 cells treated with Amwaprin exhibited a slight inhibition in their growth (Figure 2). Based on these results, a concentration of 250 ng/mL Amwaprin was chosen for further investigation, and its effect on cellular proliferation was assessed by determining the cell number for five days. The Amwaprin treatment significantly (*p* < 0.001) inhibited the proliferation of NIH-3T3 cells, with more than a two-fold decrease in proliferation observed after three days of treatment (Figure 3A). The Amwaprin treatment decreased ROS production in NIH-3T3 cells similar to its effect on growth inhibition (Figure 3B). These results reveal that Amwaprin inhibits cellular growth and ROS production in NIH-3T3 cells.

### 3.2. Amwaprin Inhibits Cytotoxicity and Apoptotic Enzyme Activity Induced via Oxidative Stress

The effect of Amwaprin on the cytotoxicity and caspase-3 activity during oxidative stress was evaluated using NIH-3T3 cells treated with H_2_O_2_. Cell viability was remarkably decreased in the absence of Amwaprin (*p* = 0.0001), whereas in the presence of Amwaprin, the viability of NIH-3T3 cells did not significantly differ after 24 or 48 h of treatment compared with the untreated cells (Figure 4A).

To determine the relationship between cytotoxicity and apoptosis regulatory enzyme activity induced by H_2_O_2_, the caspase-3 activity of NIH-3T3 cells treated with H_2_O_2_ was evaluated. Caspase-3 activity was significantly (*p* = 0.0001) decreased in NIH-3T3 cells in the presence of Amwaprin (Figure 4B), indicating that the increase in cell viability in the presence of Amwaprin was due to the inhibition of caspase-3 activity by Amwaprin.

### 3.3. Anti-Inflammatory Effect of Amwaprin

To investigate the effect of Amwaprin on the inflammatory response, the secretion of the proinflammatory mediators and cytokines IL-1β, IL-6, and TNF-α in NIH-3T3 cells was determined. The cells treated with H_2_O_2_ exhibited increased secretion of IL-1β, IL-6, and TNF-α, and treatment with Amwaprin significantly (*p* = 0.0001) decreased their secretion (Figure 5).

### 3.4. Effect of Amwaprin Binding to the Cell Membrane on Cell Protection

To further elucidate the mechanism underlying the antioxidant action of Amwaprin in NIH-3T3 cells, the interaction between Amwaprin and the cells was determined using immunofluorescence staining. Amwaprin was observed binding to the cell membrane of NIH-3T3 cells (Figure 6A). Furthermore, a similar binding mechanism was observed in Sf9 cells (Figure 6B). Subsequently, the effect of Amwaprin binding to the cell membrane on cell protection was assessed using NIH-3T3 and Sf9 cells treated with H_2_O_2_. Immunofluorescence staining revealed that Amwaprin binding to the cell membrane was associated with decreased H_2_O_2_-induced cell apoptosis. Amwaprin protects NIH-3T3 (Figure 6A) and Sf9 cells (Figure 6B) against H_2_O_2_-induced stress through direct binding to the cell membrane.

### 3.5. Radical-Scavenging Activity and DNA Protection of Amwaprin

We further confirmed that the antioxidant capacity of Amwaprin includes DPPH radical-scavenging activity (Figure 7A). The radical-scavenging assay revealed that Amwaprin exhibited a dose-dependent DPPH radical-scavenging activity compared with the controls (~17, ~22, and ~28% for 0.5, 1, and 3 μg treatments, respectively) (*p* = 0.0001), indicating the antioxidative and scavenging activities of Amwaprin against ROS. 

Based on the inhibition of H_2_O_2_-induced cell apoptosis by Amwaprin, the role of Amwaprin in defense against DNA oxidation was also investigated. The DNA nicking assay using an MCO system revealed that Amwaprin protected against hydroxyl radical DNA nicking (Figure 7B), indicating that Amwaprin plays a role in protecting DNA against ROS.

## 4. Discussion

Amwaprin, a member of the waprin family, which contains a WAP domain with eight conserved cysteine residues [21,22,23,24], was discovered in the venom of *A. mellifera* honeybees [18]. Amwaprin has been shown to have antimicrobial activity [18]. Waprins are a protein found in the venom of lower vertebrates and invertebrates, such as snakes, frogs, ants, wasps, and honeybees [18,22,23,24,25,26,27]. Apart from the antimicrobial activity of waprins, the biological action of waprins remains unknown. Here, we discovered the novel function of Amwaprin as an antioxidant and identified its underlying mechanism of action.

We discovered the inhibitory effect of Amwaprin on the growth of mammalian NIH-3T3 cells while conducting cytotoxicity assays. Amwaprin did not induce cytotoxicity but inhibited the growth and proliferation of NIH-3T3 cells. Previous studies have shown that honeybee venom inhibited the growth of colon cancer cells by inducing apoptosis [6]. Honeybee venom and melittin, a main component of honeybee venom, inhibited the growth of melanoma cells by inducing apoptosis through the upregulation of caspase-3 and caspase-9 [11]. Additionally, scorpion venom extract inhibited the growth and proliferation of tumor cells by inducing apoptosis through the upregulation of caspase-3 [32]. Based on these findings, we investigated whether Amwaprin affected caspase activity. We observed that Amwaprin inhibited cytotoxicity and caspase-3 activity in NIH-3T3 cells treated with H_2_O_2_ during oxidative stress. Since Amwaprin inhibits oxidative stress-induced cytotoxicity and caspase-3 activity, the cell growth inhibition by Amwaprin in NIH-3T3 cells may occur through mechanisms that differ from caspase-dependent apoptotic cell death, as described in previous studies using scorpion venom, honeybee venom, and melittin [6,11,32]. Meanwhile, the inhibition of cell growth and proliferation via cell cycle arrest at the G1 stage by honeybee venom [33,34] and scorpion venom [32] has been reported. However, further studies are needed to elucidate the mechanism underlying the inhibition of cell growth by Amwaprin.

Considering the increase in ROS production and cell proliferation in NIH-3T3 cells, our observations show that in the control group, ROS production due to cell proliferation was high from day 1. In contrast, the ROS levels in NIH-3T3 cells treated with Amwaprin exhibited significantly lower levels than those in the control group. This effect is likely attributed to the antioxidant activity of Amwaprin. We propose that Amwaprin may influence the inhibition of ROS production and cell growth in NIH-3T3 cells. Therefore, we conducted investigations to explore the potential association between Amwaprin and antioxidant activities. We demonstrated that during oxidative stress, the increase in cell viability in the presence of Amwaprin was due to the inhibition of caspase-3 activity by Amwaprin. Although honeybee venom typically causes inflammatory responses [5,28,35], its administration in alternative therapy exhibits anti-inflammatory effects [4,7]. H_2_O_2_ is a well-known mediator of inflammation [36,37,38,39,40]. In this study, treatments with Amwaprin significantly inhibited the secretion of the proinflammatory mediators and cytokines IL-1β, IL-6, and TNF-α in NIH-3T3 cells stimulated with H_2_O_2_, indicating that Amwaprin exhibits anti-inflammatory effects.

Considering that Amwaprin inhibits cytotoxicity, caspase-3 activity, and the secretion of proinflammatory mediators and cytokines induced by oxidative stress, it is likely that Amwaprin exhibits antiapoptotic and anti-inflammatory effects. Our findings indicate that during oxidative stress, Amwaprin inhibits the secretion of proinflammatory mediators and cytokines and reduces caspase-3 activity, consequently decreasing cytotoxicity. Moreover, Amwaprin binds to the cell membrane, and to elucidate the mechanism underlying this observation, we explored whether Amwaprin exhibits a similar mechanism of action to vitellogenin, which protects cells by directly shielding them against oxidative stress [41,42]. Notably, our data show that the binding of Amwaprin to the cell membrane exhibited a protective effect against oxidative stress-induced apoptotic cell death in Sf9 and NIH-3T3 cells, demonstrating its mechanism of action in cell protection against oxidative stress. 

We also observed that Amwaprin exhibits DPPH radical-scavenging activity and protects DNA against damage from ROS. These results suggest that Amwaprin is a scavenger of ROS, such as superoxide anion and hydroxyl radicals. Our findings indicate that Amwaprin possesses an antioxidant capacity. Compared to melittin, which exerts antioxidant and neuroprotective actions against oxidative stress [16], Amwaprin may contribute to the antioxidant function as a minor component of honeybee venom. Considering the mechanism underlying the inhibition of H_2_O_2_-induced cell apoptosis through cell membrane binding and the role of Amwaprin as a scavenger of ROS, the antioxidant capacity of Amwaprin could be attributed to the synergistic effect of radical scavenging and cell shielding. Thus, Amwaprin exhibits antioxidative and scavenging activities against ROS, reduces caspase-3 activity and H_2_O_2_-induced cell apoptosis, and protects DNA against ROS. Considering that a significant hurdle in utilizing venoms and venom peptides as possible drugs is their sometimes-non-selective toxicity towards normal cells [11,43,44,45] and given that Amwaprin has demonstrated microbicidal activity while being non-hemolytic [18], the discovery of Amwaprin’s antioxidative and cell-protective effect will offer new insights into its properties.

In previous studies, honeybee venom exhibited various anti-cancer effects, such as cytotoxicity, apoptosis, and growth arrest, in various cancer cells [46,47]. Honeybee venom induced proliferation inhibition and apoptotic death of melanoma cells [11]. Therefore, we cannot rule out the possibility that Amwaprin in honeybee venom may contribute to the proliferation inhibition of melanoma cells. However, this implication should be further validated through the inhibitory effect against various melanoma cells. Additionally, the possible effects and fundamental parameters of Amwaprin remain to be explored in the future.

## 5. Conclusions

Our findings provide the first evidence of Amwaprin’s antioxidant activity and mechanism of action, in addition to its antimicrobial activity. Amwaprin demonstrates its antioxidant function through its antiapoptotic and anti-inflammatory effects against oxidative stress by binding to the cell membrane and acting as a scavenger of ROS. Therefore, our results suggest that Amwaprin could potentially be applied as an antioxidant as well as an antimicrobial agent.

## Figures and Tables

**Figure 1 antioxidants-13-00469-f001:**
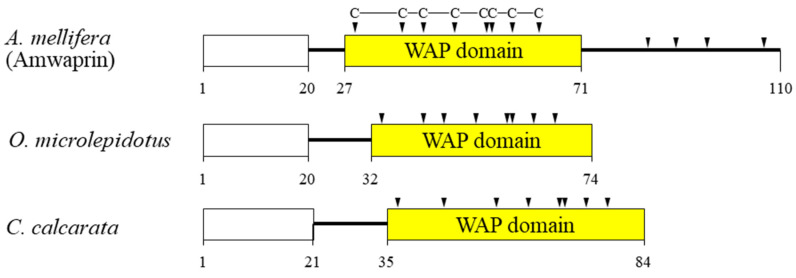
Schematic alignment of three waprins from the venom of lower vertebrates and invertebrates. The predicted signal sequence is boxed in white. The relative positions of the WAP domain (yellow) in the three waprins are shown. The conserved cysteine residues within the WAP domain are indicated by arrowheads. The aligned waprins include Amwaprin [18], *Oxyuranus microlepidotus* waprin [22], and *Ceratophrys calcarata* waprin [24].

**Figure 2 antioxidants-13-00469-f002:**
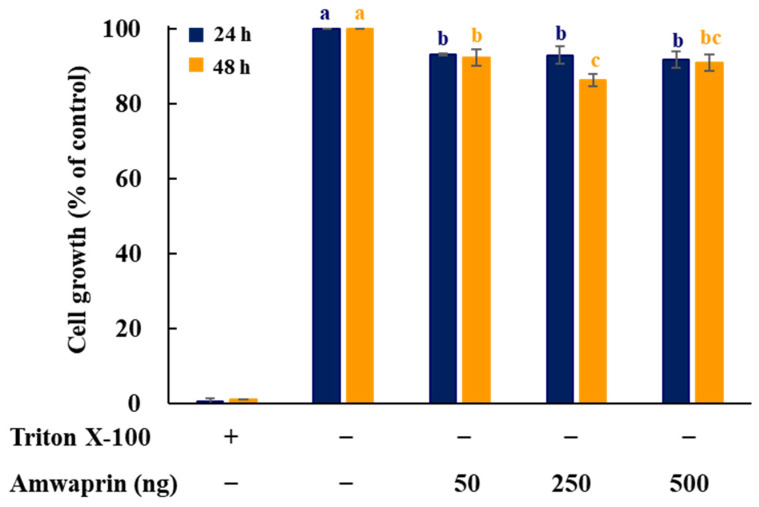
Effect of Amwaprin on the growth of mammalian cells. NIH-3T3 cells were treated with Triton X-100 (negative control) or various concentrations of Amwaprin for 24 or 48 h. Cellular growth was determined as a percentage relative to untreated cells (control). Data are represented as the mean ± SD (*n* = 3). Different letters indicate significant differences among the treatments (*p* = 0.0001).

**Figure 3 antioxidants-13-00469-f003:**
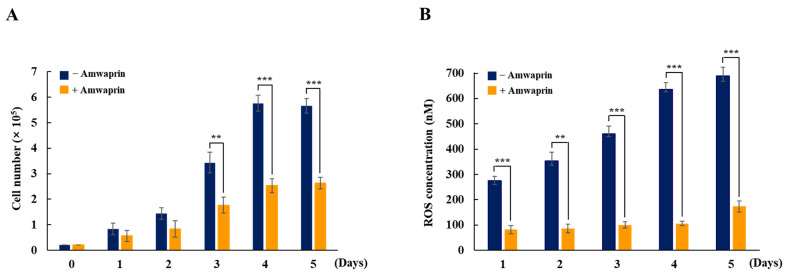
Cellular proliferation and ROS production in NIH-3T3 cells treated with Amwaprin. NIH-3T3 cells were treated with or without Amwaprin for five days, and cellular proliferation and ROS production were determined. Data are represented as the mean ± SD (*n* = 3). (**A**) Cells were counted for five days. **, *p* < 0.01 and ***, *p* < 0.001. (**B**) ROS production was measured for five days. Data are presented as ROS concentrations. **, *p* < 0.01 and ***, *p* < 0.001.

**Figure 4 antioxidants-13-00469-f004:**
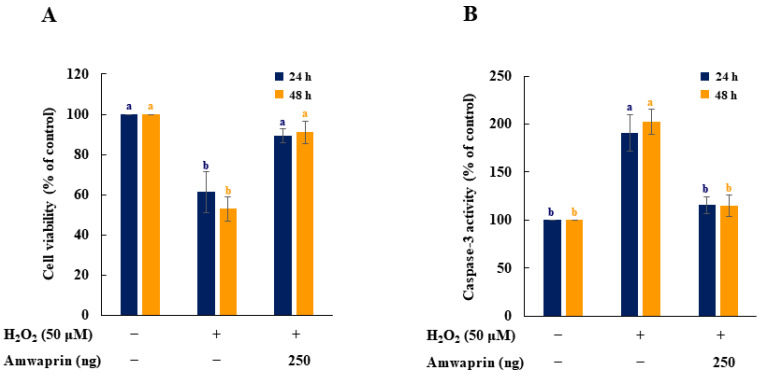
Effect of Amwaprin on cell viability and caspase-3 activity in NIH-3T3 cells. NIH-3T3 cells were treated with (+) or without (−) Amwaprin or H_2_O_2_ for 24 or 48 h. Cell viability (**A**) and caspase-3 activity (**B**) were determined as a percentage relative to the untreated cells (control). Data are represented as the mean ± SD (*n* = 3). Different letters indicate significant differences among the treatments (*p* = 0.0001).

**Figure 5 antioxidants-13-00469-f005:**
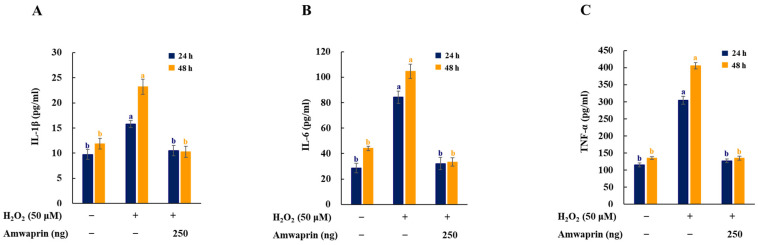
Effect of Amwaprin on the secretion of proinflammatory mediators and cytokines in NIH-3T3 cells. NIH-3T3 cells were treated with (+) or without (−) Amwaprin or H_2_O_2_ for 24 or 48 h. The concentrations of IL-1β (**A**), IL-6 (**B**), and TNF-α (**C**) were determined. Data are represented as the mean ± SD (*n* = 3). Different letters indicate significant differences among treatments (*p* = 0.0001).

**Figure 6 antioxidants-13-00469-f006:**
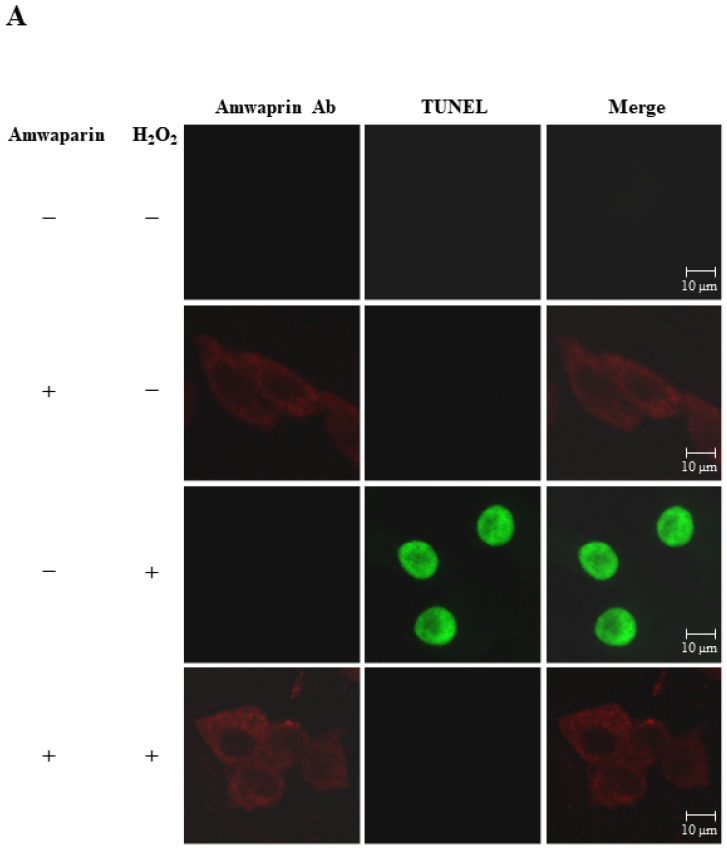
Cell protective effect of Amwaprin against H_2_O_2_-induced cell apoptosis. NIH-3T3 (**A**) and Sf9 (**B**) cells were treated with (+) or without (−) Amwaprin or H_2_O_2_. Immunofluorescence staining was performed for the detection of apoptosis (green) and Amwaprin (red) in the cells. Representative confocal images are displayed. Merged confocal images are also included (merge). Scale bar: 10 μm.

**Figure 7 antioxidants-13-00469-f007:**
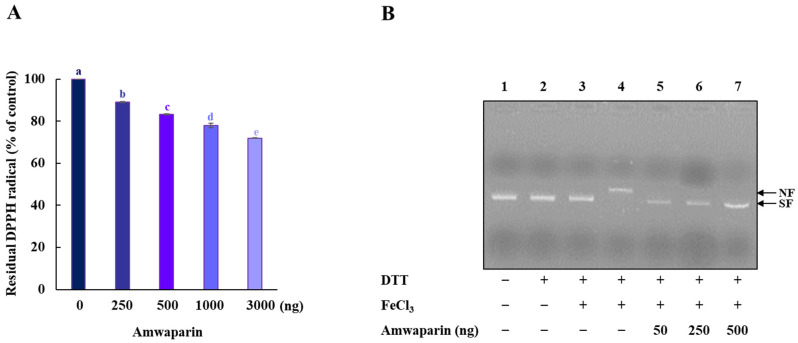
Radical-scavenging activity of Amwaprin. (**A**) DPPH radical-scavenging activity of Amwaprin. Data are represented as the mean ± SD (*n* = 3). Different letters indicate significant differences among treatments (*p* = 0.0001). (**B**) Protection of DNA cleavage by Amwaprin. Lane 1, pUC18 only; lane 2, pUC18 with DTT only; lane 3, pUC18 with FeCl_3_ only; lane 4, pUC18 with the MCO system; lanes 5–7, pUC18 with recombinant Amwaprin (50, 250, or 500 ng) in the MCO system. The nicked form (NF) and super-coiled form (SF) of the plasmid DNA are indicated by arrows.

## Data Availability

The data presented in this study are available in the article.

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
