# Peer review of "Antioxidant Activity and Mechanism of Action of Amwaprin: A Protein in Honeybee (*Apis mellifera*) Venom"

_antioxidants, 2024, doi:10.3390/antiox13040469_

Round 1

Reviewer 1 Report

The manuscript entitled “Antioxidant Activity and Mechanism of Action of Amwaprin, A Protein in Honeybee (Apis mellifera) Venom” would increase the value with the inclusion of another cell line, such as a human cell line. Also, the following comments should be discussed:

1) At page 4, line 185 the authors stated that “The Amwaprin treatment decreased ROS production in NIH 3T3 cells similar to its effect on growth inhibition (Fig. 2B).”. When observing the cell proliferation during the 5 days, the ROS production seams to increase more or less in the same proportion as the cell proliferation increase along the 5 days when comparing to day 1.

Indeed, when comparing ROS production of cells exposed to Amwaprin they increase along the 5 days (from day 1 < 100 nM to day 5 > 100nM (figure 2b). As the number of cells exposed to amwaprin are less than cells without exposure (figure 2a), for a better understanding of the ROS production, a calculation of the ROS production in relation to effective cell numbers could be helpful.

2) As the two solutions were added at the same time (Amwaprin and H2O2), there are any possible interaction between H2O2 and Amwaprin that could influence the results obtained regarding the antioxidant/anti-inflammatory potential?

Wells with only Amwaprin on the casp3 and anti-inflammatory assays could also complement results discussion.

3) Regarding immunofluorescence staining, it is unclear whether amwaprin binds to the cell membrane. In Figure 5A, amwaprin appears to be in the cytoplasm.

At figure 1 the letters a and b should be descripted.

Figure 4 caption is not below figure 4.

Triton X-100 should be considered a positive control (control that inhibit the cell proliferation). Cells without Amwaprin and/or triton are considered the negative control.

English grammar should be revised.

Author Response

Current Manuscript Number: Antioxidants-2948205

Title: Antioxidant Activity and Mechanism of Action of Amwaprin, A Protein in Honeybee (Apis mellifera) Venom

Manuscript Number: Antioxidants-2948205-R1

Dear Editor:

Thank you very much for your kind suggestions and positive consideration of our manuscript for publication. We sincerely appreciate and agree with the feedback we received. We have revised the manuscript primarily based on your suggestions and the reviewers’ comments. Below, we provide the reviewers’ comments, along with our responses. We hope that our revised manuscript meets your expectations. Thank you once again, and we look forward to hearing from you soon.

Reviewer 1

Major comments

The manuscript entitled “Antioxidant Activity and Mechanism of Action of Amwaprin, A Protein in Honeybee (Apis mellifera) Venom” would increase the value with the inclusion of another cell line, such as a human cell line. Also, the following comments should be discussed:

1) At page 4, line 185 the authors stated that “The Amwaprin treatment decreased ROS production in NIH 3T3 cells similar to its effect on growth inhibition (Fig. 2B).”. When observing the cell proliferation during the 5 days, the ROS production seams to increase more or less in the same proportion as the cell proliferation increase along the 5 days when comparing to day 1.

Indeed, when comparing ROS production of cells exposed to Amwaprin they increase along the 5 days (from day 1 < 100 nM to day 5 > 100nM (figure 2b). As the number of cells exposed to amwaprin are less than cells without exposure (figure 2a), for a better understanding of the ROS production, a calculation of the ROS production in relation to effective cell numbers could be helpful.

Author’s response;

Thank you for your valuable suggestion. Following your advice, we have included detailed information in the Discussion section. (Lines 322-331)

2) As the two solutions were added at the same time (Amwaprin and H2O2), there are any possible interaction between H2O2 and Amwaprin that could influence the results obtained regarding the antioxidant/anti-inflammatory potential?

Author’s response;

The results depicted in Fig. 5.

Discussion: Lines 355-360.

Wells with only Amwaprin on the casp3 and anti-inflammatory assays could also complement results discussion.

Author’s response;

Thank you for your valuable suggestion. Following your advice, we have included detailed information in the Discussion section. (Lines 337-342)

3) Regarding immunofluorescence staining, it is unclear whether amwaprin binds to the cell membrane. In Figure 5A, amwaprin appears to be in the cytoplasm.

Author’s response;

We observed cell membrane using confocal microscopy.

Detail comments

At figure 1 the letters a and b should be descripted.

Author’s response;

Thank you for your valuable suggestion. Following your advice, we have included comprehensive information in all figure legends.

Figure 4 caption is not below figure 4.

Author’s response;

We have revised it.

Triton X-100 should be considered a positive control (control that inhibit the cell proliferation). Cells without Amwaprin and/or triton are considered the negative control.

Author’s response;

Thank you for your valuable suggestion. Following your advice, we have included comprehensive information in the Materials and Methods section (2.2. Lines 99-101).

English grammar should be revised.

Author’s response;

Thank you for your valuable suggestion. Following your advice, we have edited the English grammar.

Reviewer 2 Report

The paper entitled “Antioxidant Activity and Mechanism of Action of Amwaprin, a Protein in Honeybee (Apis mellifera) Venom” showed that Amwaprin, a recently discovered bee venom component, has an antioxidant capacity that based on the obtained results could be attributed to the synergistic effects of radical-scavenging action and cell shielding, indicating its novel role as an antioxidant agent. This is a rather exciting new result in the field of apitherapy and in line with the various beneficial effects of bee venom constituents.

The paper entitled “Antioxidant Activity and Mechanism of Action of Amwaprin, a Protein in Honeybee (Apis mellifera) Venom” showed that Amwaprin, a recently discovered bee venom component, has an antioxidant capacity that based on the obtained results could be attributed to the synergistic effects of radical-scavenging action and cell shielding, indicating its novel role as an antioxidant agent. This is a rather exciting new result in the field of apitherapy and in line with the various beneficial effects of bee venom constituents.

Could authors provide a structural formula or even peptide sequence for the newly discovered bee venom peptide?

The authors stated that to develop effective pharmacological components, research needs to find less toxic and more effective or multifunctional natural agents. As I fully agree with the authors that the major obstacle to using venom peptides as possible drugs is their, sometimes, non-selective toxicity towards normal cells could you please comment on the toxicity of this new compound on normal non-target cells in line with the research regarding cyto/genotoxic properties of animal toxins including bee venom itself and its major constituent melittin.

Garaj-Vrhovac V, Gajski G. Evaluation of the cytogenetic status of human lymphocytes after exposure to a high concentration of bee venom in vitro. Arh Hig Rada Toksikol. 2009; 60(1): 27-34.

Sjakste N, Gajski G. A Review on Genotoxic and Genoprotective Effects of Biologically Active Compounds of Animal Origin. Toxins (Basel). 2023; 15(2): 165.

Gajski G, Domijan AM, Žegura B, Štern A, Gerić M, Novak Jovanović I, Vrhovac I, Madunić J, Breljak D, Filipič M, Garaj-Vrhovac V. Melittin induced cytogenetic damage, oxidative stress and changes in gene expression in human peripheral blood lymphocytes. Toxicon. 2016; 110:56-67.

Although referenced by previous work, could you please add more information on the recombinant Amwaprin and anti-Amwaprin antibody produced in your previous study?

Please add information from where the murine fibroblast cell line NIH 3T3 was procured.

2.3. Reactive Oxygen Species (ROS) Assay – could you please add more data regarding the protocol?

Minor remarks:

Line 188 – mind the gap (the NIH 3T3 cells)

Author Response

Current Manuscript Number: Antioxidants-2948205

Title: Antioxidant Activity and Mechanism of Action of Amwaprin, A Protein in Honeybee (Apis mellifera) Venom

Manuscript Number: Antioxidants-2948205-R1

Dear Editor:

Thank you very much for your kind suggestions and positive consideration of our manuscript for publication. We sincerely appreciate and agree with the feedback we received. We have revised the manuscript primarily based on your suggestions and the reviewers’ comments. Below, we provide the reviewers’ comments, along with our responses. We hope that our revised manuscript meets your expectations. Thank you once again, and we look forward to hearing from you soon.

Reviewer 2

Major comments

The paper entitled “Antioxidant Activity and Mechanism of Action of Amwaprin, a Protein in Honeybee (Apis mellifera) Venom” showed that Amwaprin, a recently discovered bee venom component, has an antioxidant capacity that based on the obtained results could be attributed to the synergistic effects of radical-scavenging action and cell shielding, indicating its novel role as an antioxidant agent. This is a rather exciting new result in the field of apitherapy and in line with the various beneficial effects of bee venom constituents.

Detail comments

The paper entitled “Antioxidant Activity and Mechanism of Action of Amwaprin, a Protein in Honeybee (Apis mellifera) Venom” showed that Amwaprin, a recently discovered bee venom component, has an antioxidant capacity that based on the obtained results could be attributed to the synergistic effects of radical-scavenging action and cell shielding, indicating its novel role as an antioxidant agent. This is a rather exciting new result in the field of apitherapy and in line with the various beneficial effects of bee venom constituents.

Could authors provide a structural formula or even peptide sequence for the newly discovered bee venom peptide?

Author’s response;

Thank you for your valuable suggestion. Following your advice, we have included the structure of Amwaprin with snake and frog waprins in the figure 1.

Figure 1. Schematic alignment of three waprins from the venom of lower vertebrates and invertebrates. The relative positions of the WAP domain (yellow) in the three waprins are shown. The conserved cysteine residues within the WAP domain are indicated by arrowheads. The aligned waprins include Amwaprin [18], Oxyuranus microlepidotus waprin [22], and Ceratophrys calcarata waprin [24].

The authors stated that to develop effective pharmacological components, research needs to find less toxic and more effective or multifunctional natural agents. As I fully agree with the authors that the major obstacle to using venom peptides as possible drugs is their, sometimes, non-selective toxicity towards normal cells could you please comment on the toxicity of this new compound on normal non-target cells in line with the research regarding cyto/genotoxic properties of animal toxins including bee venom itself and its major constituent melittin.

Garaj-Vrhovac V, Gajski G. Evaluation of the cytogenetic status of human lymphocytes after exposure to a high concentration of bee venom in vitro. Arh Hig Rada Toksikol. 2009; 60(1): 27-34.

Sjakste N, Gajski G. A Review on Genotoxic and Genoprotective Effects of Biologically Active Compounds of Animal Origin. Toxins (Basel). 2023; 15(2): 165.

Gajski G, Domijan AM, Žegura B, Štern A, Gerić M, Novak Jovanović I, Vrhovac I, Madunić J, Breljak D, Filipič M, Garaj-Vrhovac V. Melittin induced cytogenetic damage, oxidative stress and changes in gene expression in human peripheral blood lymphocytes. Toxicon. 2016; 110:56-67.

Author’s response;

Thank you for your valuable suggestion. Following your advice, we have included previous papers in the Discussion (Lines 360-364) and References (44-46).

Although referenced by previous work, could you please add more information on the recombinant Amwaprin and anti-Amwaprin antibody produced in your previous study?

Author’s response;

We have included comprehensive information in the Materials and Methods section (2.1).

Please add information from where the murine fibroblast cell line NIH 3T3 was procured.

Author’s response;

We have included comprehensive information in the Materials and Methods section (2.2).

2.3. Reactive Oxygen Species (ROS) Assay – could you please add more data regarding the protocol?

Author’s response;

We have included comprehensive information in the Materials and Methods section (2.3).

Minor remarks:

Line 188 – mind the gap (the NIH 3T3 cells)

Author’s response;

We have revised it.

Round 2

Reviewer 2 Report

Authors made substantial changes warranting application.

Authors answered all the raised comments and question and changed their paper accordingly.